

# Marsupial and monotreme milk—a review of its nutrient and immune properties

Hayley J. Stannard[1], Robert D. Miller[2] and Julie M. Old[3]

[1] School of Animal and Veterinary Sciences, Charles Sturt University, Wagga Wagga, NSW, Australia
[2] Center for Evolutionary and Theoretical Immunology, Department of Biology, University of New Mexico, Albuquerque, NM, USA
[3] School of Science, Western Sydney University, Penrith, NSW, Australia

## ABSTRACT

All mammals are characterized by the ability of females to produce milk. Marsupial (metatherian) and monotreme (prototherian) young are born in a highly altricial state and rely on their mother's milk for the first part of their life. Here we review the role and importance of milk in marsupial and monotreme development. Milk is the primary source of sustenance for young marsupials and monotremes and its composition varies at different stages of development. We applied nutritional geometry techniques to a limited number of species with values available to analyze changes in macronutrient composition of milk at different stages. Macronutrient energy composition of marsupial milk varies between species and changes concentration during the course of lactation. As well as nourishment, marsupial and monotreme milk supplies growth and immune factors. Neonates are unable to mount a specific immune response shortly after birth and therefore rely on immunoglobulins, immunological cells and other immunologically important molecules transferred through milk. Milk is also essential to the development of the maternal-young bond and is achieved through feedback systems and odor preferences in eutherian mammals. However, we have much to learn about the role of milk in marsupial and monotreme mother-young bonding. Further research is warranted in gaining a better understanding of the role of milk as a source of nutrition, developmental factors and immunity, in a broader range of marsupial species, and monotremes.

# INTRODUCTION

Mammals evolved approximately 200 million years ago (*Flight, 2011*) and today comprise three groups—prototherians, metatherians and eutherians. Monotremes (prototherians) diverged from marsupials (metatherians) and eutherians approximately 166 million years ago (*Renfree et al., 2009*), and marsupials and eutherians separated around 130 million years ago (*Bininda-Emonds et al., 2007*; *Luo et al., 2003*; *Nilsson et al., 2010*). Monotremes lay eggs and their young hatch in a highly altricial state. Marsupials, however, give birth to live young in a highly altricial state and most of their development is completed after birth, whilst eutherians give birth to young varying from altricial to precocial.

Corresponding author
Julie M. Old,
j.old@westernsydney.edu.au

When compared to eutherians, newborn marsupials have been described as similar to a gestationally eight week old human fetus (*Block, 1964*), hence much of the development of marsupials occurs in the external environment. Thus, when comparing the development of marsupials and eutherians, the time of "pouch emergence" of marsupial young is often regarded as similar to the time of birth in eutherians. However, despite these differences in developmental stages at birth, all three groups of mammals are characterized by the ability of the female of the species to produce milk. Via the mammary gland, milk supplies growth factors and immunological components to young mammals. Immediately or hatching after birth, milk is the sole form of sustenance. Mammalian young rely on their mother to provide all the nutrients for their initial growth and development. As mammals are diverse in their characteristics, size and habitats that they live in, milk composition varies markedly depending on the needs of the young. This article specifically reviews the crucial role of milk in young marsupial (metatherian) and monotreme (prototherian) development with a focus on both nutrition and immunology. Furthermore, we highlight other largely neglected areas of research where milk is essential to development such as its role in maternal-young bonding and where further research is required. This review will be of particular interest to those conducting research in mammalogy, nutritional geometry, immunology, mammary gland biology and lactation.

## SURVEY METHODOLOGY

Scientific journal articles were sourced for this review by conducting searches using the GoogleScholar® and Web of Science® databases. The search strategy used the combined terms "milk and monotreme" or "milk and marsupial". We screened and filtered the articles, limiting those with relevant titles and written in English. We identified further relevant articles referenced in the articles found. Generally, we included articles written no more than 30 years ago, although some earlier articles were incorporated due to their importance in this review to provide further clarity and enhancement.

### Mammary glands and pouches

Mammary glands are specialized accessory glands of the skin (*Cowie, 1974*) and are similar to sweat glands in their secretory process, having evolved from apocrine-like skin glands (*Lefèvre, Sharp & Nicholas, 2010*). Mammary glands evolved prior to the radiation of today's extant mammals (reviewed in *Lefèvre, Sharp & Nicholas, 2010*; *Power & Schulkin, 2013*), around 166–240 million years ago when the first mammals arose (*Bininda-Emonds et al., 2007*).

Both eutherians and marsupials have nipples or teats to aid transfer of milk to the young, whilst echidnas (*Tachyglossus* sp. and *Zaglossus* spp.) and the platypus (*Ornithorhynchus anatinus*) have the ducts of a single mammary gland opening directly onto the areola in association with a hair follicle, and young monotremes (puggles) suckle milk from the pores of these milk patches (*Griffiths, McIntosh & Coles, 1969*). The size and shape of mammary glands differs between species. The number of mammary glands, although usually occurring in pairs (*Tyndale-Biscoe & Renfree, 1987*), differs between

species. Among marsupials, there is generally an inverse relationship between body size of the female and the number of teats, with small species such as phascogales and bandicoots having eight teats (*Stannard et al., 2013*), and larger species such as wombats and koalas having two, and macropods having four (*Tyndale-Biscoe, 2005*). The gray short-tailed opossum (*Monodelphis domestica*) is an extreme example having 13 mammary glands (*Tyndale-Biscoe, 2005*).

The position of mammary glands (and teats) also varies. In marsupials with pouches, such as red kangaroos (*Macropus rufus*), the teats are located within a pit on the dorsal surface of the pouch, and each teat has a separate mammary gland (*Griffiths, McIntosh & Leckie, 1972*). In marsupials the number of mammary glands, hence the number of teats, also determines the maximum litter size as marsupial young permanently attach to the maternal teat shortly after birth. The mammary gland subsequently grows correspondingly in size with the growth of the pouch young, as does the teat, in length (*Griffiths, McIntosh & Leckie, 1972*). The number of teats for each species represents the maximum number of young that can be sustained for each litter. Thus in some dasyurid marsupials the transit to the maternal teat and attachment can lead to a fight for survival, as some marsupials give birth to supernumerary young (*Bradley, 1997*; *Gemmell, Veitch & Nelson, 2002*; *Nelson & Gemmell, 2003*), and more young are born than there are teats.

The mammary gland tissue consists primarily of glandular tissue with alveoli and ducts amongst a stroma of adipose and connective tissue (Cowper's ligaments) (*Geddes, 2007*). While marsupials lack the suspensory ligaments associated with eutherians (*Cowie, 1974*), the ilio-marsupialis muscle of the marsupial mammary glands and teats enable the mother to draw their young close to the body, in a similar manner to the cremaster muscle in males that draws the testes closer to the body (*Griffiths & Slater, 1988*). Male South American water opossums or yapok (*Chironectes minimus*), an aquatic marsupial, have a pouch that is used as a watertight compartment where they house the scrotum (*Hunsaker, 1977*), and was likely similar to that of the now extinct Tasmanian tiger (*Thylacinus cynocephalus*) that had a backward facing pouch to also enclose their scrotum (*Beddard, 1891*). *Woolley et al. (2002)* have suggested in the later stages of lactation that the ilio-marsupialis muscles could be used to shake off the suckling young.

There is increasing evidence of the role of cells associated with the immune system in the normal development and function of marsupial mammary tissue. T cells infiltrate the mammary tissue early in the first post-natal week in the gray short-tailed opossum and transcriptome analyses are consistent with the majority being γδ T cells (*Fehrenkamp & Miller, 2019b*). What role these γδ T cells are playing in mammary protection or the protection of the suckling young is not known. Also noteworthy is the appearance of eosinophils early in the postnatal period when the mammary tissue is undergoing significant remodeling in the opossum. The presence of eosinophils correlates with local expression of the cytokine interleukin-16, a known chemotactic factor for eosinophils (*Fehrenkamp & Miller, 2019a*). It has been speculated that in eutherians eosinophils play a role in mammary remodeling, whether the same is true in marsupials is unknown, but this appears to be an ancient conserved feature in mammals.

## Odor identification and communication between mother and young

Although nipples and teats in marsupials and eutherians provide a delivery mechanism for milk to young mammals, they also provide a sensory attractant, including touch, olfaction and taste, for young mammals to locate milk. The same is true for milk patches in monotremes. The suckling in marsupials and eutherians by the young in turn induces oxytocin release in the mother, inducing milk letdown and maintenance (*Lincoln & Renfree, 1981*), as well as inducing maternal behaviors (*Dwyer, 2014*).

Olfaction is not only essential for newborn mammals to locate the maternal nipple and feed, but also important in kin recognition and maternal-young bonding. In rabbits (*Oryctolagus cuniculus*) the mammary pheromone has been shown to influence sucking performance and hence milk intake in newborn kittens (*Jouhanneau, Schaal & Coureaud, 2016*). Mammary pheromone is secreted by the nipples of lactating rabbits, however the precise source is unknown (*Moncomble et al., 2005*). It may be produced by the skin glands at the tip of the nipple and dissolved in milk, secretions of sebaceous and apocrine glands within the nipple dermis secreted into the galactophorous ducts, or compounds produced or changed by microflora in the galactophorous ducts (*Moncomble et al., 2005*). The olfactory systems of marsupials and monotremes are still immature at birth/hatching, however given their developmental stage it is likely young can use olfactory cues to find their way to their mother's milk (*Schneider, 2011*). Behavioral testing by *Schneider et al. (2009)*, for example, have shown that neonate tammar wallabies (*Notamacropus eugenii*) use olfactory cues to locate their mother's odor. *Nishitani et al. (2009)* also found human young show reduced pain responses when exposed to the odor of their mother's milk, but a reduction in pain response whilst suckling is yet to be investigated in marsupials or monotremes. Therefore, although no studies have been conducted on the odors produced by the marsupial teat (and pouch), odors likely aid the newborn marsupial to locate the teat and attach, as well as aid development of the young-mother bond.

## Nutrient composition of milk

The ability to produce milk varying in composition at the same time is called asynchronous concurrent lactation (*Nicholas, 1988*). Some marsupials, such as the tammar wallaby, are able to produce milk with varying compositions at the same time, whereby one mammary gland supplies a newborn permanently attached to one teat, whilst a second mammary gland supplies milk with a different composition to an older young at foot. Similarly, the red kangaroo can produce two different kinds of milk, depending on the age of the young (*Griffiths, McIntosh & Leckie, 1972*). One gland can produce a fluid rich in protein for a neonate and the other "mature" milk for a young at foot (*Griffiths, McIntosh & Leckie, 1972*). Asynchronous concurrent lactation is possible due to changes in the extracellular matrix composition of the mammary gland, and these changes subsequently regulate milk protein expression (*Wanyonyi et al., 2013*). In marsupials, milk composition and yield also vary with sex of the young (*Quesnel et al., 2017*; *Waterman, Robert & Braun, 2012*). Both tammar wallabies and eastern grey kangaroos allocate more protein to sons than to daughters (*Quesnel et al., 2017*; *Waterman, Robert & Braun, 2012*).

Milk consists of milk solids (made up of proteins, lipids, carbohydrates and minerals) and water. Throughout lactation pronounced changes occur to the nutrient composition which is a characteristic of marsupial milk and makes it different to that of eutherian milk (*Green, 1984*; *Green, Newgrain & Merchant, 1980*). In monotremes, while there is very limited information available, it likewise appears that their milk composition also changes over time (*Grant, Griffiths & Leckie, 1983*; *Griffiths et al., 1984*; *Teahan, McKenzie & Griffiths, 1991*).

Depending on the species, differences in milk composition produced at different times occur during early, mid and late lactation, but can also be referred to as different phases or stages. It is difficult to make comparisons between species when discussing these different phases/stages/time of lactation because the definitions can vary widely between species, and even in the literature for the same species. Therefore, throughout this paper where possible we refer to the phase/stage/time of lactation for the species we are discussing as defined in the literature for that species. For example, early lactation (sometimes referred to as Phase 2) consists of the milk produced from birth until the end of fixation, where young are no longer permanently attached to the teat (*Tyndale-Biscoe & Janssens, 1988*). In some species, such as the tammar wallaby, phase 2 can also be separated into 2A and 2B. Phase 2A represents the period where the young remains permanently attached to the teat, and 2B represents the stage whereby the young start to intermittently detach (*Pharo, 2019*). Late lactation (Phase 3) begins when the milk becomes similar to that of eutherian milk, and the young is no longer permanently attached to the teat (*Adamski & Demmer, 2000*), and includes the time of permanent pouch exit, and when the young starts to eat solid foods (*Tyndale-Biscoe & Janssens, 1988*), but the timing of teat detachment, or end of fixation or fixed period of lactation, and pouch exit differs greatly and is not consistently defined for each species. In addition, the timing of the lactation phases differs for each species, for example phase 2A is 0–125 days in the tammar wallaby and 0–70 days in the quokka (*Setonix brachyurus*) (*Green, Newgrain & Merchant, 1980*; *Miller, Bencini & Hartmann, 2009*). Early lactation protein, whey acidic protein and late lactation proteins are expressed exclusively during early, mid and late lactation and also act as markers for the different phases (*Nicholas et al., 1997*), but have not been studied in all species, and hence may differ among different species. For these complex reasons, we will refer to early, mid and late lactation loosely, and not defined by the literature for each species due to its variation in meaning.

In early lactation (soon after parturition) milk solids in marsupial milk are relatively low and the milk is dilute, in later stages of lactation the milk becomes thicker. Milk solids increase as lactation progresses (*Green, 1984*; *Green, Newgrain & Merchant, 1980*; *Ikonomopoulou, Smolenski & Rose, 2005*; *Nicholas, 1988*). Milk solids can be as low as 9% weight for weight (w/w) in early lactation and increase to 54% w/w in late lactation depending on species (*Crowley, Woodward & Rose, 1988*; *Green, Krause & Newgrain, 1996*). In some marsupials, including the Virginian opossum (*Didelphis virginiana*), common brushtail possum (*Trichosurus vulpecula*), common ringtail possum and koala (*Phascolarctos cinereus*), milk solids peaking in early to mid-lactation prior to pouch exit (*Cowan, 1989*; *Green, Krause & Newgrain, 1996*; *Krockenberger, 1996*; *Munks et al., 1991*).

*Krockenberger (1996)* suggested that more dilute milk may provide an important water source for young koalas prior to weaning, however it is possible mothers produce dilute milk to conserve their own energy resources. Monotreme milk is rich in solids, with 48.9% w/w in short-beaked echidna (*Tachyglossus aculeatus*) milk and 39.1% w/w in platypus milk (*Griffiths et al., 1984*). Similar to marsupial milk, monotreme milk appears to be more dilute in early lactation with platypus having 8% solids (*Grant, Griffiths & Leckie, 1983*) and short-beaked echidnas 12% (*Griffiths, McIntosh & Coles, 1969*), however data are based on small samples sizes.

### Protein

Total macronutrient quantities change throughout lactation and qualitative changes also occur to the macronutrients. Protein concentration increases throughout lactation in most marsupials (*Ikonomopoulou, Smolenski & Rose, 2005*; *Merchant et al., 1996*; *Rose, Shetewi & Flowers, 2005*). By comparison, protein content either remains stable or only decreases slightly in eutherian species, for example as seen in rhesus monkeys (*Macaca mulatta*), insectivorous bats and equids (*Kunz et al., 1995*; *Lönnerdal et al., 1984*; *Oftedal & Jenness, 2009*). Casein and whey protein increase gradually in tammar wallaby milk (*Nicholas, 1988*), with pre-albumin and α-globulin fractions increasing from 180 days postpartum and peak at pouch exit (*Green & Renfree, 1982*). Levels of serum albumin, α-lactalbumin, β-lactoglobulin and transferrin are relatively consistent throughout lactation in the red kangaroo (*Muths, 1996*; *Nicholas et al., 2001*). However, whey acidic protein secretion begins at around day 155 post-partum and ceases at day 209, cystatin is secreted from day 102 until day 192, and late lactation protein A is secreted from day 192 (*Nicholas et al., 2001*). In the quokka whey albumin, γ-globulin and β-globulin levels remain consistent throughout lactation and are present at levels lower than maternal serum albumin (*Jordan & Morgan, 1968*). Common brushtail possum transferrin increases around the time of pouch exit, as much as ten-fold between days 110 and 130 (*Grigor et al., 1991*).

Casein and α-globulin proteins are the main proteins in milk, including monotreme milk. Overall, caseins are quite different across mammalian species, however most species have no more than four different casein proteins (*Ginger & Grigor, 1999*). Initially platypus were found to only have one casein protein while short-beaked echidnas had two (*Grant, 2007*; *Marston, 1926*; *Teahan, McKenzie & Griffiths, 1991*), and more recently *Lefèvre, Sharp & Nicholas (2009)* characterized the caseins in both species. The whey proteins of the short-beaked echidna are distinctly different from those of the platypus (*Joseph & Griffiths, 1992*). Total protein and casein composition of echidna milk is high in early lactation (until day 48) (*Teahan, McKenzie & Griffiths, 1991*).

### Lipids

Lipids increase gradually in a range of marsupial milks during early lactation, and then increase exponentially into late lactation and reach a peak around weaning (*Crowley, Woodward & Rose, 1988*; *Green, Merchant & Newgrain, 1987*; *Ikonomopoulou, Smolenski & Rose, 2005*; *Muths, 1996*). Crude lipid concentration can increase by up to five

times in late lactation compared to early lactation (*Rose, Shetewi & Flowers, 2005*). In three arboreal folivores lipid concentration peaks in mid-lactation. In koalas this occurs just prior to pouch exit, before reducing in late lactation (*Cowan, 1989*; *Krockenberger, 1996*; *Munks et al., 1991*). The common ringtail and brushtail possums have the lowest concentrations of crude lipid (maximum 8 g/100 mL) in their milk compared to other marsupials studied thus far (*Cowan, 1989*; *Munks et al., 1991*). By comparison lipid concentration can be as high as 46 g/100 g in the eastern quoll (*Dasyurus vivverinus*) or 31.4 g/100 g in the Tasmanian pademelon (*Thylogale billardierii*) (*Green, Merchant & Newgrain, 1987*; *Rose, Shetewi & Flowers, 2005*).

Oleic acid (CI8:1) followed by palmitic acid (CI6:0) are the predominant fatty acids present in marsupial milk (*Crowley, Woodward & Rose, 1988*; *Green, 1984*; *Green, Merchant & Newgrain, 1987*; *Grigor, 1980*). In the eastern quoll palmitic acid is higher than oleic acid in early lactation and they crossover between 6 and 10 weeks post-partum with oleic acid becoming the dominant fatty acid (*Green, Merchant & Newgrain, 1987*). Stearic acid (CI8:0) and linoleic acid (CI8:2) are also major fatty acids in marsupial milk (*Crowley, Woodward & Rose, 1988*; *Grant, Griffiths & Leckie, 1983*; *Green, Merchant & Newgrain, 1987*; *Griffiths et al., 1988*; *Griffiths, McIntosh & Leckie, 1972*; *Poole et al., 1982*), as well as palmitolic acid (C16.1) in long-nosed potoroo (*Potorous tridactylus*) milk (*Crowley, Woodward & Rose, 1988*). Arachidonic acid (C20:4) is present in eastern quoll and numbat (*Myrmecobius fasciatus*) milk more so than in tammar wallaby milk, likely due to the insectivorous diet of quolls and numbats (*Green, Merchant & Newgrain, 1987*; *Griffiths et al., 1988*). Short-chain fatty acids are not found in the long-nosed potoroo, tammar wallaby or numbat milk while only trace amounts are found in eastern grey and red kangaroo milk (*Crowley, Woodward & Rose, 1988*; *Green, Griffiths & Leckie, 1983*; *Griffiths et al., 1988*; *Griffiths, McIntosh & Leckie, 1972*; *Poole et al., 1982*).

Monotreme milk shows some diversity in fatty acid composition across species. Short-beaked echidna milk is low in polyunsaturated fats compared to platypus milk which is high in polyunsaturated fats, such as linolenic acid (C18:3) (*Augee, Gooden & Musser, 2006*; *Griffiths, 1978*; *Griffiths et al., 1973*). In addition, short-beaked echidnas lack very long-chained polyunsaturated fats such as docosahexaenoic acid (C22:6), docosatetraenoic acid (22:4) and docosapentaenoic acid (C22:5) (*Griffiths et al., 1973*). Oleic acid (C18:1) is the dominant fatty acid in monotreme milk, with short-beaked echidnas having more oleic acid than platypuses (*Griffiths et al., 1984*). The fatty acids in platypus milk are the same as those found in the food of platypuses (macroinvertebrates) (*Gibson et al., 1988*). Fatty acid composition is influenced by captivity and diet in short-beaked echidnas, with significant differences observed between oleic and palmitic acids in captive and free-ranging short-beaked echidna milk (*Augee, Gooden & Musser, 2006*). Similar to marsupials and monotremes, eutherian pinnipeds also show increases in lipid content of milk during lactation (*Arnould & Hindell, 1999*; *Riedman & Ortiz, 1979*). For example, northern elephant seal (*Mirounga angustirostris*) lipid levels increase from around 12% to 50% during lactation (*Riedman & Ortiz, 1979*). This increase occurs as water content decreases.

Lipid content in milk has been suggested to be linked to nursing patterns, whereby those mothers being absent from their young for long periods produce higher quantities of fat, and those mothers that are always close to their young and therefore able to feed their young more frequently have lower levels of fats but higher carbohydrate levels (*Oftedal, 2002*; *Sharp et al., 2006*). This is also true of some seal species that leave their young for periods of time to go out and forage (*Dosako et al., 1983*; *Riedman & Ortiz, 1979*). Marsupial neonates are permanently attached to the maternal teat and marsupial milk contains low levels of fat during this period (*Crowley, Woodward & Rose, 1988*; *Green, Merchant & Newgrain, 1987*). The lipid levels increase as the young grows and detaches from the teat prior to weaning (*Crowley, Woodward & Rose, 1988*; *Green, Merchant & Newgrain, 1987*; *Ikonomopoulou, Smolenski & Rose, 2005*; *Muths, 1996*). It has been suggested the lipid increases in marsupial milk can be correlated to the need for fat reserves required for thermoregulation and increased muscle activity related to locomotion in the young (*Merchant et al., 1996*; *Munks et al., 1991*).

### Carbohydrates

Carbohydrate concentration in marsupial milk gradually increases post-partum, followed by a sharp decrease around the time of pouch exit. For example, in the tammar wallaby total hexose peaks at 13% at 26 weeks and falls to 1% by 40 weeks pospartum (*Messer & Green, 1979*; *Messer et al., 1982*), and a similar pattern is observed in the red-necked wallaby (*Macropus rufogriseus*) and red kangaroo (*Merchant et al., 1989*; *Muths, 1996*). In the eastern quoll, milk carbohydrate concentration peaks at eight weeks postpartum and rapidly decreases from 17 weeks post-partum prior to the onset of weaning at 20–22 weeks (*Messer et al., 1987*). Carbohydrates also undergo qualitative changes in composition throughout lactation that, like lipids and proteins, are timed to correspond to the developmental needs of the developing young undergoing the equivalent of "fetal" development outside the womb.

Lactose is the main carbohydrate in eutherian milk, however this is not the case in monotremes and marsupials (*Bergman & Housley, 1968*; *Jenness, Regehr & Sloan, 1964*; *Messer & Kerry, 1973*). There are differences between carbohydrate composition across marsupial species. In the tammar wallaby carbohydrates change from lactose to oligosaccharides between days 4 and 21 days post-partum (*Messer, Griffiths & Green, 1984*). In the gray short-tailed opossum early lactation milk contains monosaccharides (galactose and glucose) as well as lactose, and after two weeks the milk carbohydrate changes to predominantly oligosaccharides that coincides with the end of fixed lactation (*Crisp, Messer & VandeBerg, 1989*). During late lactation of the gray short-tailed opossum, milk contains lactose, oligosaccharides and monosaccharides, while tammar wallaby milk contains monosaccharides as the only carbohydrates (*Crisp, Messer & VandeBerg, 1989*; *Messer & Green, 1979*). In the potoroo early milk is mostly oligosaccharides and then changes to lactose, galactose and glucose at 15 weeks post-partum (*Crowley, Woodward & Rose, 1988*). Arboreal folivore milk has similar compositional changes in carbohydrates. In the common brushtail possum lactose is the main carbohydrate, with traces of higher oligosaccharides in early lactation (*Crisp, Cowan & Messer, 1989*). During mid-lactation

the carbohydrate portion of koala milk is dominated by oligosaccharides, while during late-lactation it is dominated by lactose, similar to the common ringtail and brushtail possums (*Crisp, Cowan & Messer, 1989*; *Krockenberger, 1996*; *Munks et al., 1991*).

Monotreme milk has a high fucose composition and the principal carbohydrate in echidna milk is sialyl-lactose, and difucosyl-lactose in the platypus whereas these are only minor carbohydrates in human milk (*Messer, 1974*; *Messer & Kerry, 1973*). In Tasmanian short-beaked echidnas (*T. a. setosus*) difucosyl-lactose was only detected in late lactation (~150 days post hatching) (*Oftedal et al., 2014*). Lactose is present in only trace amounts during early lactation in short-beaked echidna milk (*Oftedal et al., 2014*). It is hypothesized that fucose is the main energy source for young monotremes and is comparable to galactose and glucose in eutherian mammals (*Messer et al., 1983*). Sialyl-lactose found in monotreme milk is unique to the echidna and not found in any other mammal milk (*Messer, 1974*; *Oftedal et al., 2014*).

### Minerals (Electrolytes)

Comparatively less data are available on electrolytes compared to macronutrients in marsupial and monotreme milk. Sodium and potassium levels are variable. Usually sodium is found in higher concentrations than potassium in early lactation. There is an inverse correlation with these two minerals in late lactation when potassium levels become higher than sodium levels (*Cowan, 1989*; *Merchant et al., 1989*; *Merchant et al., 1996*). In the red-necked wallaby, red kangaroo, and common ringtail and brushtail possums sodium levels increase around the time of weaning just prior to lactation ceasing (*Cowan, 1989*; *Merchant et al., 1989*; *Munks et al., 1991*; *Muths, 1996*). Atypically American marsupials show little change in sodium or potassium concentrations over the course of lactation (*Green, Krause & Newgrain, 1996*; *Green, Vandeberg & Newgrain, 1991*). Sodium and potassium concentrations of monotremes (means from 37 to 99 days postpartum in the short-beaked echidna and unknown ages of platypus (*Griffiths et al., 1984*)) are within the range observed in the brush-tailed bettong (*Bettongia penicillata*) in early lactation (day 55) (*Merchant, Libke & Smith, 1994*).

Generally iron concentrations show a decreasing trend during lactation in marsupial and eutherian mammals (*Loh & Kaldor, 1973*). Comparatively marsupial and monotreme milk contain higher levels (>3x) of iron than eutherian species (*Green, Merchant & Newgrain, 1987*; *Griffiths, 1983*; *Kaldor & Ezekiel, 1962*). Marsupials and monotremes rely on iron supplied in milk for hemoglobin, cytochromes, and myoglobin due to their premature state at birth and prolonged need for milk (*Griffiths, 1983*). Due to the underdeveloped stage of marsupial livers at birth they are unable to store as much iron as eutherians and thus require higher levels from milk. Iron concentration in the milk can be up to nine times the maternal plasma levels in marsupials (*Kaldor & Morgan, 1986*).

In the common ringtail possum no clear trends for calcium and magnesium concentrations are evident (*Munks et al., 1991*). In the red kangaroo calcium peaks in mid-lactation and reduces after pouch emergence, whilst magnesium concentrations are low relative to other electrolytes in the milk (*Muths, 1996*). In the Virginian opossum calcium peaks in early lactation and stays high until 10 weeks post-partum and decreases

in late lactation (*Green, Krause & Newgrain, 1996*). Calcium and magnesium concentrations of monotremes are similar to that in marsupials, however data are means (from 37 to 99 days postpartum in the short-beaked echidna), and from unknown ages of platypus (*Griffiths et al., 1984*), thus changes in these mineral concentrations cannot be determined. In an insectivorous eutherian, the nine-banded armadillo (*Dasypus novemcinctus*), calcium concentration increases (~x2.9) throughout lactation between 3 and 6 days postpartum and 49–51 days postpartum (*Power et al., 2018*), so it is possible monotremes undergo similar increases to those of marsupials and eutherians however more research is required.

### Macronutrient energy

We applied nutritional geometry techniques to published data to assess the nutrition of marsupial milk. Using the Atwater factors, protein 17 kJ/g, lipid 37 kJ/g, carbohydrate 16 kJ/g (*Merrill & Watt, 1973*), composition of milk on a dry matter basis published data were converted into a percentage energy basis and graphed to determine the percentage of macronutrient energy contribution to milk at different times during lactation and comparisons made across two species (Table S1). In the tammar wallaby and eastern quoll protein contributes a less variable percentage of energy than carbohydrate and lipid, remaining between 17% and 28% for both species (Fig. 1). In early lactation for both species carbohydrate contributes a greater percentage of energy. Later in lactation closer to weaning time, lipid contributes the most energy in milk of both species (Fig. 1). Higher lipid energy would support young at a time of accelerated growth and development and potential aid in fat storage.

## Milk as sustenance for young

Growth of young is related to the quantity and quality of the milk produced and thus higher intake leads to faster growth rates. In marsupials the body mass of young range from 10 to 750 mg at birth (*Tyndale-Biscoe, 2005*), thus milk intake in early lactation is low and increases exponentially. For example, quokka pouch young increase milk intake from 1.6 mL d$^{-1}$ at 55 days postpartum to 32.5 mL d$^{-1}$ at 165 days postpartum (*Miller, Bencini & Hartmann, 2010*). Marsupial young grow at a rate of 0.2 to 0.5 g per mL of milk consumed depending on species and age (*Green, Merchant & Newgrain, 1988*; *Merchant, Libke & Smith, 1994*; *Merchant et al., 1996*; *Miller, Bencini & Hartmann, 2010*; *Munks & Green, 1997*; *Smolenski & Rose, 1988*). The tammar wallaby grows at a rate of 0.21–0.25 g mL$^{-1}$ during the first 24 weeks postpartum and then increases to 0.37 g mL$^{-1}$ after 25 weeks (*Green, Merchant & Newgrain, 1988*). By comparison the brush-tailed bettong grows at a rate of 0.51 g mL$^{-1}$ at 4–6 weeks postpartum and then decreases to 0.40 g mL$^{-1}$ at 13 weeks (*Merchant, Libke & Smith, 1994*). Differences in growth rates during lactation are likely related to changes in energy composition of the milk throughout lactation and other factors (see section on "Macronutrient Energy"). As well as total milk intake, *Smolenski & Rose (1988)* suggested that protein intake from milk caused differences in growth rates between two similar-sized marsupial species.

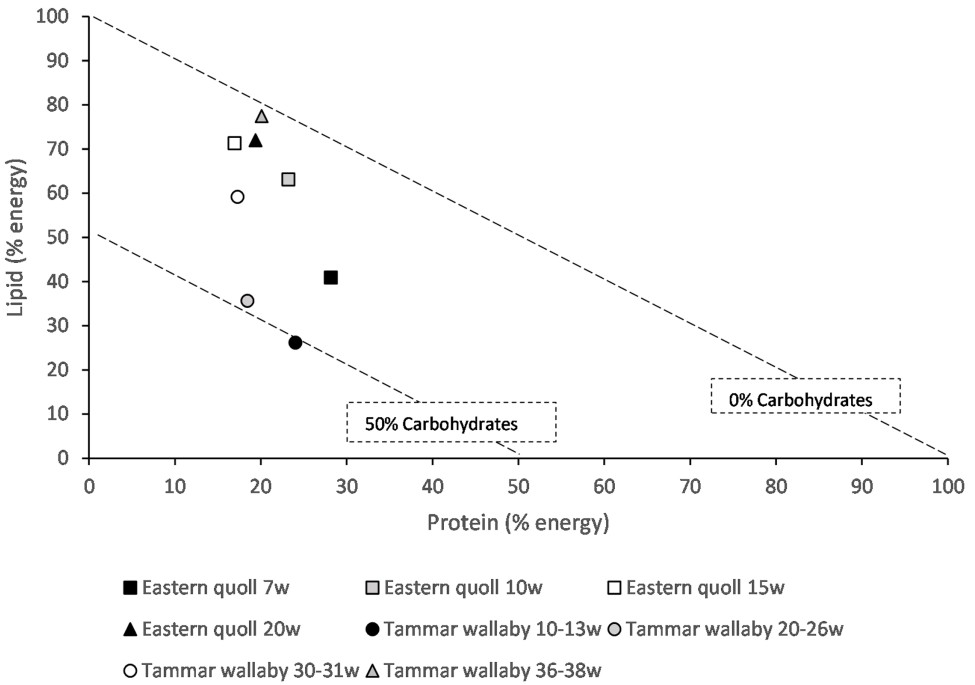

**Figure 1** **Right angle mixture triangle of macronutrient composition of milk from the eastern quoll and tammar wallaby at different weeks (w) of lactation.** For the tammar wallaby values represent phase 2A, 2B and two time points within phase3. Protein and lipid (% energy) increase along the *X* and *Y* axes, respectively. The percentage of carbohydrate decreases with distance from the origin, with the diagonal lines representing a fixed percentage carbohydrate.

Short-beaked echidna young can suckle 10–20% of their body mass in 30–60 min (*Green, Griffiths & Newgrain, 1985*; *Griffiths, 1965*). As lactation progresses frequency of feeding young decreases in monotremes and the time the mother spends out foraging for food increases (*Morrow & Nicol, 2012*; *Rismiller & McKelvey, 2009*; *Thomas et al., 2020*). *Green, Griffiths & Newgrain (1985)* estimated that echidna young grow at a rate of 0.41 ± 0.10 g mL$^{-1}$ of milk consumed.

## Maternal nutrition

Lactation is energetically costly for all mammals with energy needs ranging from 35% to 149% above maintenance costs for mothers (*Gittleman & Thompson, 1988*; *Hayssen, 1993*). Marsupials have a long lactation period where significant changes occur to milk composition, and for marsupial young, most growth occurs at this time. Thus, marsupial mothers invest a significant proportion of energy towards milk production over a longer period of time than comparatively-sized eutherian mammals, whereas eutherians invest more in gestation. In the marsupials it appears that energetic needs increase significantly in late lactation by 112–222% (*Atramentowicz, 1992*; *Cork, 1991*; *Krockenberger, 2003*; *Loudon, 1987*; *Stannard & Old, 2015*). Changes to maternal energetic needs correspond to milk becoming more energy-rich and containing a higher lipid concentration. In the eastern quoll 9–14 weeks postpartum, there is an energetic increase to 200% that of non-lactating animals that coincides with an increase in milk production

(*Green, Merchant & Newgrain, 1997*). During gestation and early lactation there are minimal increases in the energetic needs of marsupial mothers (*Atramentowicz, 1992*; *Cork, 1991*; *Loudon, 1987*). However, restricting maternal nutrition affects the nutritional composition of milk produced by marsupial mothers (*Green & Merchant, 1988*). For example, bettongs fed on a restricted diet in captivity produced milk with a lower protein content compared to those on an *ad lib* diet and free-ranging animals (*Rose et al., 2003*). Access to lower quality forage caused eastern grey kangaroos to produce lower-energy milk than mothers in a previous year that were also in better body condition (*Quesnel et al., 2017*).

Some marsupial mothers contribute very little fat reserves towards lactation and rely on current nutrient intake during lactation (*Cork, 1991*; *Krockenberger, 2003*; *Loudon, 1987*; *Stannard & Old, 2015*). In the wild, this has implications on survival of young, for example if mothers cannot access the required nutrition for themselves, as well as their young, they may have to abandon their young to be able to survive. One example is that red kangaroos have a high mortality of young-at-foot and pouch young during times of drought and poor quality food availability (*Frith & Sharman, 1964*). When resources are plentiful, marsupials that have multiple young at once will produce milk of the similar quality/quantity for their litter no matter how many young they have, causing young in smaller litters to grow faster (*Atramentowicz, 1992*; *Stannard & Old, 2015*) and potentially wean earlier. For the bare-tailed wooly opossum (*Caluromys philander*) having one young, as opposed to eight, does not require a significant increase in energy intake, and thus energy intake by mothers with one pouch young is no different to non-lactating females (*Atramentowicz, 1992*). Northern brown bandicoots (*Isoodon macrourus*), and common ringtail and brushtail possums appear to use body stores to cope with late lactation, as well as increase their energy intake, similar to other marsupials (*Bell, 1981*; *Merchant, 1990*; *Munks & Green, 1995*). The northern brown bandicoot stores body reserves in the first 30–40 days of lactation then uses these stores in the latter half of lactation (*Merchant, 1990*). It is suggested these stores are fat that can be mobilized when the mothers require them for late lactation (*Merchant, 1990*). In the common ringtail possum the body stores are possibly fat (*Munks & Green, 1995*), however this has not been investigated. Eastern grey kangaroos use both stored and incoming resources to cope with the cost of lactation (*Gelin et al., 2016*; *Quesnel et al., 2017*). There is some indication that short-beaked echidnas use body stores to support lactation (*Griffiths, 1965*; *Morrow & Nicol, 2012*). *Morrow & Nicol (2012)* found echidna mothers do not leave the nursery burrow to feed during the first few weeks after the young have hatched, and subsequently lost significant body mass, however their body mass increased again after leaving the burrow and recommencing feeding. Platypus will consume up to 2,093 kJ kg$^{-1}$ d$^{-1}$ during lactation which is 227% that of non-lactating females (*Thomas et al., 2020*). Like marsupials, there seems to be minimal increases in energetic needs of female platypus in the first month of lactation (*Thomas et al., 2017*; *Thomas et al., 2020*). More data is needed on maternal nutrition from a wider range of marsupials and monotremes to understand how nutrient demands change throughout lactation.

## Growth factors in milk

Milk not only provides essential nutrition required for development but also plays a role in signaling growth and development in young (*Donovan & Odle, 1994*; *Joss et al., 2009*; *Kobata et al., 2008*; *Waite et al., 2005*). The components that are not essential for nutrition but used in cell and tissue growth are known as bioactives, and include proteins, peptides, fatty acids, hormones and growth factors (*Donovan & Odle, 1994*). The reader is referred to *Sharp et al. (2014)* and *Sharp et al. (2017)* for comprehensive reviews of bioactives in milk of marsupials and monotremes. *Modepalli et al. (2015)* and *Modepalli et al. (2016)* found milk collected from tammar wallabies likely aids lung maturation. *Modepalli et al. (2014)* also identified miRNA in tammar wallaby milk and found they act as markers for lactation and mammary gland development in the mother and may aid development of the young. Tammar wallaby milk contains exosomes that are likely to be able to cross the neonatal gut and hence enter the bloodstream (*Daly et al., 2007*; *Old & Deane, 2003*), therefore allowing miRNA within the milk to impact growth and development (*Sharp et al., 2014*). Further, cross fostering studies in marsupials have shown that when a younger marsupial neonate is placed onto a teat secreting later-stage milk, the neonate will develop faster (*Kwek et al., 2009*; *Waite et al., 2005*), which is possibly due to the differences in bioactives supplied in milk of different stages as well as a higher energy content (*Green & Merchant, 1988*).

Earlier studies of cross-fostering in marsupials, particularly cross-species, were trialed over 60 years ago (*Merchant & Sharman, 1966*), because of its potential as a conservation tool, to remove the need for the donor species to continue lactation and support of the newly born young, and allow subsequent mating, thus enhancing population numbers through additional births (*Druery et al., 2007*; *Finlayson et al., 2007*). Most studies have since been conducted on macropods (*Clark, 1968*; *Johnson, 1981*; *Jones, Temple-Smith & Taggart, 2004*; *Merchant & Sharman, 1966*; *Schultz, Whitehead & Taggart, 2006*; *Taggart et al., 2005*), but other studies have investigated bettongs (*Smith, 1998*) and wombats (*Finlayson et al., 2007*). Interspecies cross-fostering has had mixed success. Success has been reported for pouch young transferred from wild bridled nailtail wallabies (*Onychogalea fraenata*) and spectacled hare wallabies (*Lagorchestes conspicillatus*) to captive unadorned rock wallaby (*Petrogale inornata*) mothers (*Johnson, 1981*), and for pouch young transferred from wild brush-tailed rock wallaby (*Petrogale penicillata*) and yellow-footed rock wallaby (*Petrogale xanthopus*) mothers to captive tammar wallaby mothers (*Taggart, Schultz & Temple-Smith, 1997*). However, *Johnson (1981)* has also documented mortality and retarded growth after transfer of wild whiptail wallaby (*Macropus parryi*) pouch young to agile wallaby (*Macropus agilis*) mothers, and *Menzies et al. (2007)* has noted similar impacts on pouch young after transfer from parma wallabies (*Macropus parma*) to tammar wallabies. *Finlayson et al. (2007)* has since suggested synchrony of reproductive cycles, in both target and recipient species, is essential for cross-fostering success.

A further protein (S100A19) has been identified in tammar wallaby neonatal forestomach and expressed only when the neonate is exclusively digesting milk

(*Kwek et al., 2013*). The same protein is expressed in the tammar wallaby mammary gland tissue but is only expressed during pregnancy and involution (*Kwek et al., 2013*). It is believed S100A19 is involved in gut development or plays an antibacterial role in both the neonatal gut and the mammary gland (*Kwek et al., 2013*). A more recent and comprehensive microarray study conducted on tammar wallaby mammary gland expression at different lactation periods has found, despite many immunological-related proteins being expressed during crucial developmental periods (*Joss et al., 2009*), that the molecular expression of some genes were down regulated, and some genes varied greatly in their levels of expression, such as *SIM2* and *FUT8*, however their function in lactation is yet to be determined (*Vander Jagt et al., 2016*).

## Milk replacers

Milk replacers provide an alternative to maternal milk to raise young when the mother cannot. The nutrient composition of milk varies among different species as discussed earlier. For that reason, milk from one species is generally not suitable for another. Milk replacers have been used for a variety of marsupials and a monotreme, including bare-nosed wombats (*Vombatus ursinus*), Tasmanian devils (*Sarcophilus harrisii*) and the short-beaked echidna (*Jackson, 2003*). Marsupials may require hand-raising when a mother rejects their young, is unable to produce enough milk, or has died. It is recommended that marsupials be fed milk with very low levels, or no lactose and galactose, hence eutherian milk is not suitable. There are a range of commercially available milk replacers that have been designed specifically to meet the needs of marsupials and monotremes that should be used.

Using data available on commercially available milk replacers, we mapped macronutrient energy using the Atwater factors, protein 17 kJ/g, lipid 37 kJ/g and carbohydrate 16 kJ/g. On a percentage energy basis milk replacers adequately replicate carbohydrate and protein energy for most marsupial milks during mid-lactation but they do not contain as much lipid as available in maternal milk (Fig. 2). Generally, the milk replacers recommended for late lactation have higher lipid concentrations. In addition to being used to raise young, milk replacers have been used as supplements for older underweight animals, specifically koalas and bilbies (*Macrotis lagotis*), to help get animals back into condition (*Gamble & Blyde, 1992*).

## Milk aids immunological defense of the young

As discussed previously marsupials and monotremes are born relatively under-developed compared to eutherian mammals. Newborn marsupials have been compared to an eight week old human fetus in terms of their developmental stage (*Block, 1964*), and are unable to mount specific immune defense (*Old & Deane, 2000*). At the time of birth, the young marsupial crawls from the vagina to the teat and permanently attaches. On attachment the teat swells in the neonate's mouth and the mouth fuzes around the teat. As the neonate is unable to mount a specific immune response shortly after birth, and due to a lack of circulating mature lymphocytes (reviewed in *Old & Deane (2000)*), the young marsupial is believed to be highly reliant on maternally-derived strategies for immune defense, including immunoglobulins transferred prenatally via the yolk-sac in

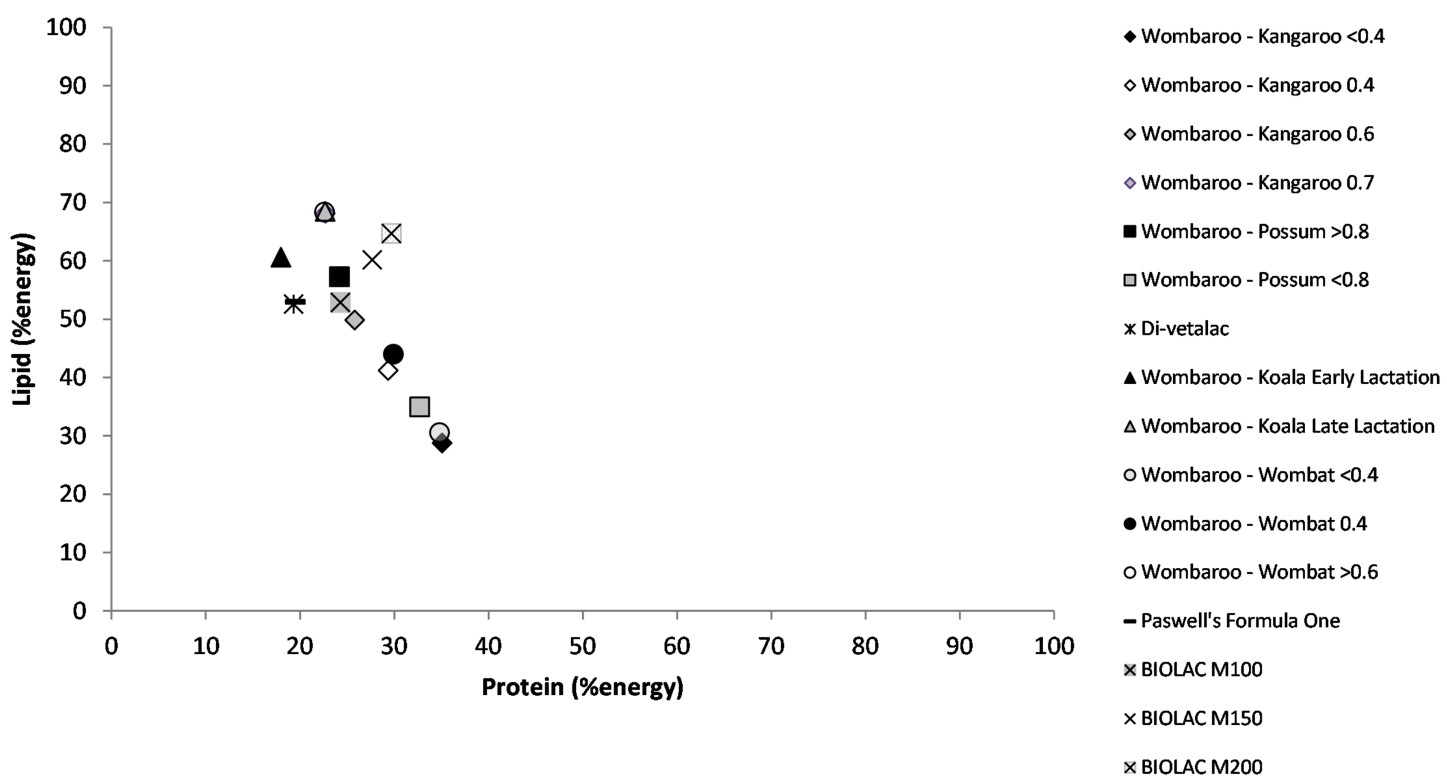

**Figure 2  Right-angled mixture triangle of commercially available marsupial milk formulas.** Protein and lipid (% energy) increase along the *X* and *Y* axes, respectively. The percentage of carbohydrate decreases with distance from the origin.

some species (*Deane, Cooper & Renfree, 1990*), immunological cells (*Young et al., 1997*), and other immunologically important molecules in milk (*Joss et al., 2009*; *Lefèvre et al., 2007*).

Neutrophils in all stages of development have been found in tammar wallaby milk and were the predominant cell type in colostrum and early stage milk of tammars and long-footed potoroos (*Potorous longipes*) (*Young et al., 1997*), and late stage milk collected from koalas (*Young & Deane, 2001*). In contrast, lymphocytes were observed in early and late stage tammar milk, but not in colostrum samples, and macrophages were only found in late stage tammar and yellow-footed rock-wallaby (*Petrogale xanthopus*) milk (*Young et al., 1997*). *Basden, Cooper & Deane (1996)* found high levels of circulating neutrophils in newborn tammar wallabies and these may have been maternally-derived given the numbers in milk. *Cockson & McNeice (1980)* also suggested this method of maternally-derived transfer of cells occurred across the neonatal quokka gut, as they noted small and large lymphocytes in the milk supplied to young 12 days postpartum and neutrophils in the milk of young 20 days postpartum.

Transplacental immunoglobulin(Ig) transfer although shown to occur in the tammar wallaby (*Deane, Cooper & Renfree, 1990*), does not occur in all marsupials, as evidenced by a lack of circulating Ig in newborn Virginian (*Hindes & Mizell, 1976*) and gray short-tailed opossums (*Samples, Vanderberg & Stone, 1986*). Ingested Igs have been detected in the serum of common brushtail possums (and quokka) in the switch phase of lactation

(up until at least 98 days), but not in the late phase of lactation (150 days) (*Yadav, 1971*), however no pouch young younger than 40 days postpartum were examined. Further, although the concentration of IgG in colostrum is low in the tammar wallaby and hill kangaroo (*Macropus robustus*), after suckling the level of IgG in the serum of the young dramatically increases, suggesting passive uptake of IgG from the colostrum (*Deane, Cooper & Renfree, 1990*; *Deane & Cooper, 1984*).

Early milk contains colostrum, a protein-rich substance with high levels of Igs, as well as proteins important for transport, nutrition and immune protection (*Joss et al., 2007*). IgA has been detected in the early phase of tammar wallaby (*Deane, Cooper & Renfree, 1990*) and common brushtail possum (*Adamski & Demmer, 1999*) lactation. Expression of IgA, the J-chain, responsible for the dimeric formation of IgA, and polymeric Ig receptor (PIGR), the molecule responsible for transferring the IgA complex through epithelial cells, have been reported in the common brushtail possum mammary gland at all stages of lactation. However IgA and J-chain expression was heightened during the first week of lactation and towards the end of the switch phase, when the young marsupial first leaves the pouch, and PIGR expression was similar in its expression levels except that it remained high after the end of the switch phase until lactation ended (*Adamski & Demmer, 1999*). Thus, IgA appears important for immune protection of the young during the early neonatal period and at the time of pouch exit (*Adamski & Demmer, 1999*). Further, *Adamski & Demmer (2000)* reported two distinct stages during common brushtail possum lactation when immune transfer was increased, these being the early lactation and switch phases. Expression of FCRN, responsible for regulating IgG in milk, is also highest during the early lactation phase and decreases over time, whereas, β-2-microglobin increases during the switch phase when IgG secretion from the mammary gland is highest (*Adamski, King & Demmer, 2000*).

More recently, *Fehrenkamp, Morrissey & Miller (2019)* demonstrated a correlation between FcRN expression in the mammary of lactating gray short-tailed opossums with FcRN expression in the gut of the neonates. As expression declined, presumably to reduce maternal IgG transfer as the young matured, there was concomitant reduction in expression in the gut of the suckling young. The timing of this decline correlates also with the maturity of the neonatal immune system and the appearance of endogenous IgG in the neonate (*Wang, Sharp & Miller, 2012*). These observations are consistent with reducing transfer of maternal IgG timed with the newborn developing its own ability to respond to pathogens and generate active protective immunity and less dependance on passive maternal immunity. Furthermore, there is little evidence of resident, IgG producing B cells in opossum mammary tissue, suggesting that the IgG being transferred is from maternal circulation. In contrast there are resident IgA producing B cells in the opossum mammary (*Fehrenkamp, Morrissey & Miller, 2019*).

As stated earlier in this review total protein was highest in the late phase of lactation in the tammar wallaby (*Green, Newgrain & Merchant, 1980*; *Green & Renfree, 1982*). Proteins of primary immunological importance include Igs, myeloid cathelicidin and complement component B, whilst other components may provide secondary levels of protection such as very early lactation protein (VELP) (*Joss et al., 2009*). VELP has also

been identified in common brushtail possum (*Kuy et al., 2007*) and koala (*Johnson et al., 2018*) milk. VELP may play a role against bacterial colonization of the mucosal surfaces of the gastrointestinal tract (*Joss et al., 2007*). MM1 has also recently been identified in the koala genome and suggested to have antimicrobial properties (*Johnson et al., 2018*). Other proteins are involved in nutrition, for example β-lactoglobin, and development such as retinal A and α-fetoprotein (*Joss et al., 2009*).

Monotremes express a unique monotreme lactation protein (MLP) in a variety of tissues including milk, with the highest expression occurring in milk cells (*Enjapoori et al., 2014*). Analysis of the recombinant protein shows it is N-glycosylated, amphipathic and α-helical, and therefore typical of an antimicrobial protein (*Enjapoori et al., 2014*; *Newman et al., 2018*). Functional studies have confirmed the recombinant protein has antimicrobial properties against *Staphylococcus aureus* and a commensal bacteria, *Enterococcus faecalis* (*Enjapoori et al., 2014*). Further antimicrobial proteins have also been found in echidna (EchAMP) and platypus (PlatAMP) milk (*Bisana et al., 2013*; *Enjapoori et al., 2014*), and shown to have bacteriostatic properties against Gram positive and negative bacteria but not *Enterococcus faecalis* (*Bisana et al., 2013*).

Cathelicidins, a type of antimicrobial peptide, found in most vertebrates are also produced by marsupials (*Carman et al., 2009*, *2008*; *Johnson et al., 2018*) and have been identified in milk (*Daly et al., 2008*; *Wanyonyi et al., 2011*). *Warren et al. (2008)* have also identified cathelicidins (*Whittington et al., 2008a*, *2008b*) and defensins in the platypus genome, and *Whittington et al. (2009)* in the tissues (brain, kidney, liver, lung, spleen and testis), but did not find cathelicidins or defensins in platypus milk.

*Nicholas et al. (1997)* described the changes in composition of whey proteins in tammar wallaby milk throughout lactation and correlated these with the different stages of lactation. *Joss et al. (2009)* also described the protein components of whey over the four lactation periods (phase 1 (early), 2A and 2B (switch phase) and 3 (late)) of the tammar wallaby and correlated the changes in protein content to the changes in developmental stages of the newborn marsupial. Whey acidic protein four- disulfide domain protein-2, for example, is increased in pregnancy, high in phase 2A, downregulated in phase 2B, upregulated again at the end of phase 3 and high in involution (*Watt et al., 2012*), and has been shown to have antibacterial activity against a range of pathogenic bacteria but not *Enterococcus faecalis*. The increased levels of expression at these times correlate to the time at which the mammary gland is most at risk of infection (*Daly et al., 2007*). The time of increased expression is also likely to provide the young with additional immune protection whilst they are developing immunocompetence and unable to mount their own immune specific response (*Old & Deane, 2000*).

Studies investigating gene expression in tammar wallaby mammary gland tissue using expressed sequence tags has found increased expression of genes over the period of lactation, particularly secreted proteins (*Lefèvre et al., 2007*). These proteins were involved in translation, and metabolic and immune functions (*Lefèvre et al., 2007*). *Piotte et al. (1997)* also confirmed the presence of lysozyme, an antimicrobial enzyme, and found its protein structure to be similar to bovine stomach lysozymes and primate lysozymes, but

different to calcium-binding lysozymes in the echidna (*Kikuchi, Kawano & Nitta, 1998*; *Teahan et al., 1991*) and horse (*McKenzie & Shaw, 1985*).

## CONCLUSIONS

Milk is essential for marsupial and monotreme sustenance and development. Nutrient changes in marsupial milk are very pronounced. Although data on nutrient changes throughout lactation is available for some marsupial species, there is limited data available for monotremes. Additional information on the changes in macronutrient composition of monotreme milk would allow for a better understanding of those features that are ancient and conserved among mammals versus those that are unique to individual mammalian lineages.

Noteworthy, from the literature, are the changes that maternal energy requirements undergo during late lactation, when maternal energy requirements can increase by 222% above basal. The increased energy needs correspond to when milk becomes more lipid rich. Maternal food intake and sex of offspring correlates to differences in milk macronutrient concentration.

Commercially available milk replacers provide a substitute for maternal milk. Replacers have been developed that replicate the macronutrient composition of marsupial and monotreme milk for captive rearing of young. What remains lacking is a greater understanding of milk composition at different stages of development for specific species. Such will aid conservation of those species through provision of milk more closely resembling that of mothers throughout development.

Whilst a good body of research exists for the role of milk in some specific marsupials, further research is required to address the role of a range of factors present in milk and their direct effect on newborns from a broader range of marsupial and monotreme species. It is also clear that the wide and varied definitions of the different stages and phases of milk production that it hinders comparison between many of the studies on marsupials. Conservation of marsupials and monotremes would also benefit from furthering our understanding of their species-specific biology.

## ACKNOWLEDGEMENTS

The authors would like to thank Professor Kevin Nicholas for helpful discussions and suggestions on an earlier draft.

### Funding
The authors received no funding for this work.

### Competing Interests
The authors declare that they have no competing interests.
## Author Contributions

- Hayley J. Stannard conceived and designed the experiments, performed the experiments, analyzed the data, prepared figures and/or tables, authored or reviewed drafts of the paper, and approved the final draft.
- Robert D. Miller performed the experiments, authored or reviewed drafts of the paper, and approved the final draft.
- Julie M. Old conceived and designed the experiments, performed the experiments, analyzed the data, authored or reviewed drafts of the paper, and approved the final draft.

## Data Availability

All data are contained within the article and the Supplemental File.

## Supplemental Information

Supplemental information for this article can be found online at http://dx.doi.org/10.7717/peerj.9335#supplemental-information.

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
