# Peer review of "Marsupial and monotreme milk—a review of its nutrient and immune properties"

_PeerJ, doi:10.7717/peerj.9335_

## Round 0.1 · original submission · Minor Revisions

The reviewers commented on the thoroughness of your review but suggested that some sections can be condensed. This would also allow some detail to be added as suggested for section 4 by reviewer 2. The authors should ensure the paper is properly formatted and in accordance with PeerJ. For example section 5 appears before section 4.1.

Reviewer 1 ·

Basic reporting

The review is an exhaustive up to date overview of the diversity of studies performed on marsupial and monotreme milk and what is known so far on its nutrient and immune properties. This represents a valuable contribution to lactation research at this time.

Experimental design

No comment

Validity of the findings

No comments

Additional comments

Minor correction suggestions:

ref 8 line 70, should be instead ref 78 where the apocrine-like gland origin of the mammary gland was originally proposed: Olav T. Oftedal, The Mammary Gland and Its Origin During Synapsid Evolution. Journal of Mammary Gland Biology and Neoplasia volume 7, pages225–252(2002)
I also suggest writing "apocrine-like" rather than just "apocrine"

line 177: this should be plural "but have not been studied..." as the refers to 3 lactation specific proteins. Alternatively, perhaps better "but this has not been studied.." if referring to their use as markers.

line 200: I suggest "peaking" instead of "peak".

line 324: Atwater factors should be expressed in energy per gram: "kJ/g", not "kJ". see also line 450.

line 332: this is more likely "at a time of accelerated growth..." rather than "is a time of accelerated growth...".

line 519: "may provide" rather than "may to provide"

line 581: the sentence needs adjustment, perhaps "It is also clear that the wide and varied definitions of the different stages and phases of milk production hinders comparison between many of the studies on marsupials".

·

Basic reporting

Marsupial and monotreme milk - A review of its nutrient and
immune properties


A comprehensive and very long description of the nutrient properties of marsupial and monotreme milk. The authors are to be complimented on the comprehensive nature of this well organized and well-written manuscript. It is a timely review will provide a valuable resource for years to come.

Some suggestions for improvement

The lengthy descriptive nature impeded an understanding of the importance of milk components, especially during the early stages of development. For example section 4.3. “Carbohydrates” – Why do carbohydrates change? This was explained well for iron and maternal nutrition. Without substantially increasing the manuscript, some interpretation of the biological significance of the milk components as different stages would be beneficial

Section 11 Conclusions. This was the weakest link in this manuscript. Rather than conclusions, it was more of an abbreviated review. Statements such as “Further research is needed to assess changes in macronutrient composition of monotreme milk.” need to be justified and explained why more research is needed. Is it simply for the sake of knowledge or does it have conservation issues?

A suggestion would be to restructure the conclusions away from description to interpretation and significance. This could be linked to conservation and artificial milk formulations. There are almost 200 references, but “further research is needed” is often stated. It would be beneficial to indicate why. Is the research invalid, wrong species, wrong nutrients etc.?

Referencing used an incorrect format as PeerJ uses the 'Name. Year' style with an alphabetized reference list, not numbered as used in this review. The reference section was also incorrect.

Some comments have been added directly to the pdf document.

Experimental design

Not relevant as this was a review article.

Validity of the findings

No comment.

---

## Round 0.2 · accepted · Accept

The revision addresses all the concerns of the two reviewers.